# Automated exploitation of the big configuration space of large adsorbates on transition metals reveals chemistry feasibility

Geun Ho Gu [1,2✉], Miriam Lee[3], Yousung Jung [2✉] & Dionisios G. Vlachos [3✉]

Mechanistic understanding of large molecule conversion and the discovery of suitable heterogeneous catalysts have been lagging due to the combinatorial inventory of intermediates and the inability of humans to enumerate all structures. Here, we introduce an automated framework to predict stable configurations on transition metal surfaces and demonstrate its validity for adsorbates with up to 6 carbon and oxygen atoms on 11 metals, enabling the exploration of ~$10^8$ potential configurations. It combines a graph enumeration platform, force field, multi-fidelity DFT calculations, and first-principles trained machine learning. Clusters in the data reveal groups of catalysts stabilizing different structures and expose selective catalysts for showcase transformations, such as the ethylene epoxidation on Ag and Cu and the lack of C-C scission chemistry on Au. Deviations from the commonly assumed atom valency rule of small adsorbates are also manifested. This library can be leveraged to identify catalysts for converting large molecules computationally.

[1] School of Energy Technology, Korea Institute of Energy Technology, 200 Hyuksin-ro, Naju 58330, South Korea. [2] Department of Chemical and Biomolecular Engineering, Korea Advanced Institute of Science and Technology (KAIST), 291 Daehak-ro, Daejeon 34141, South Korea. [3] Department of Chemical and Biomolecular Engineering and Catalysis Center for Energy Innovation (CCEI), University of Delaware, Newark, DE 19716, USA. ✉email: ggu@kentech.ac.kr; ysjn@kaist.ac.kr; vlachos@udel.edu

The advancement in density functional theory (DFT) has enabled mechanism development and in silico catalyst design[1]. DFT calculations have been performed for several small-molecule chemistries, including hydrogen evolution and oxidation reactions[2,3], oxygen reduction and evolution reactions[4–8], $CO_2$ reduction, $N_2$ reduction[9,10], and $CH_4$ activation[11]. Computing the species configurations and thermochemistry is essential, as correlated uncertainty quantification reveals that more thermodynamic parameters than activation energy parameters affect the kinetics[12]. Adsorbate configurations are prerequisites in computing activation energies of elementary reactions. While manual DFT calculations have been adequate for small molecules, they are impractical for large molecules due to the combinatorial size of the reaction network that includes all intermediates[13]. Thus, an extension of computations to large molecules on transition-metal catalysts has been lagging. Establishing a framework for modeling large molecules would thus be essential to significantly accelerate mechanistic and discovery studies, for example, in renewable energy, such as biomass pyrolysis and gasification[14,15], biomass upgrade via hydrogenation[16–18] and hydrodeoxygenation[19,20], and hydrogen production via biomass reforming[21], and recycling of plastics.

Surprisingly, the challenge in DFT calculations of adsorbates is not merely the computational cost–databases in the order $10^6$ are becoming commonplace[22–25]. A challenge is the automated generation of stable adsorbate configurations on surfaces. The adsorption configuration of large molecules is combinatorially intractable to enumerate in practice due to the multiple adsorption sites and several surface-binding atoms[26]. Each stable configuration can undergo different chemistry, and the reaction network thus depends critically on identifying all (or at least the most) stable configurations. It turns out that this task escapes intuition.

Several tools can ease the generation of stable configurations. Peterson et al.[27] developed a global adsorbate configuration optimization method using the constrained minima hopping method, but its scalability is limited as DFT-based annealing is used. Medford and coworkers utilized the minima hopping with faster density functional theory tight-binding (DFTB) methods for bidentates, but obtaining reliable DFTB parameters is not trivial[28]. Bligaard and coworkers have implemented graph-based enumeration for bidentate adsorbate configurations[29], and Greeley and coworkers developed a python-based graph theory package to encode the adsorption structure into a graph to identify the adsorption structures and generate high coverage configurations uniquely[30]. Currently, no general strategy exists that systematically identifies stable adsorbate configurations with three or more surface-binding atoms needed to adequately describe the chemical reactions of large molecules on metal surfaces.

Here, we introduce a general framework to predict a nearly complete set of stable adsorbate configurations on metal surfaces. We introduce expert knowledge-based enumeration rules to generate the configuration space, containing most, if not all, stable configurations. The configurations are optimized using a force field, and strained configurations are removed. For the configurations with ≤3 heteroatoms (non-hydrogen organic atoms), we perform multi-fidelity DFT calculations to assess the configuration stability. With this data, we train a machine learning (ML) model and use it as a screening tool to predict the stability of larger adsorbates before performing DFT calculations. The workflow is summarized in Fig. 1. We apply the framework to close-packed surfaces of Ag, Au, Co, Cu, Ir, Ni, Pd, Pt, and discover 4,979 stable configurations. The predictive ability of the ML model-based screening is further demonstrated for 1650 configurations with 4 ≤ heteroatoms ≤ 6 also computed via DFT. We find that distinct trends in stable configurations among catalysts explain the observed selectivity in experimental systems,

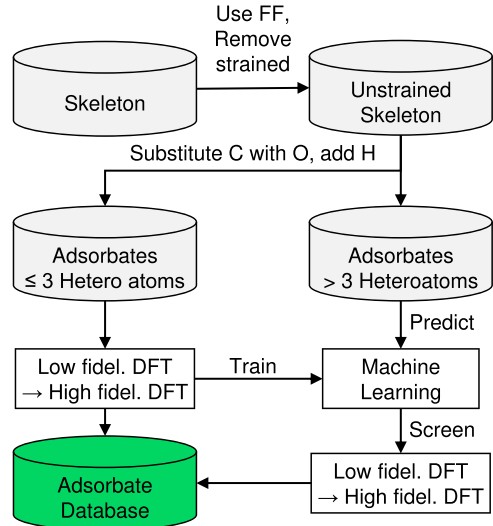

**Fig. 1 Workflow for the automated identification of stable adsorption configurations in the huge chemical space.** Skeleton configurations containing only the carbon are enumerated exhaustively using the knowledge-based rules, followed by force field (FF) optimization to remove strained configurations. An unstrained skeleton is converted to adsorbates by substituting O and adding hydrogen atoms. For adsorbates with ≤3 heteroatoms (non-hydrogen atoms), multi-fidelity DFT calculations are performed to assess their stability. The model is used to predict the stability of larger adsorbates. Based on the score, screening selects the promising candidates for DFT calculations. The stable adsorbates are gathered to make the final set.

and the clustering in the adsorbate data is rationalized by the d-band/adsorbate interactions. We propose that stable intermediates are essential for a catalyst to carry out a specific reaction, and the extensive library created here can be leveraged to pre-screen catalysts for all commonly metal-catalyzed chemistries. This work paves the foundation toward mechanistic insights into and design principles of large molecule conversion.

## Results

**Skeleton enumeration.** We introduce graph transformation rules to enumerate "skeleton" configurations, which contain carbons and their connectivity patterns to the surface, inspired by Ruddigkeit et al.[31] The initial pool of configurations are built by adding carbon on top, bridge, and hollow sites on a large surface lattice graph. Then, the rules that precisely add one carbon atom are repeatedly applied to build larger configurations. Hydrogen additions and electronic effects are considered later.

Four types of rules can comprehensively enumerate all possible adsorbate configurations. The first type adds an adsorbed carbon to an adsorbed carbon (surface propagation rules). These rules can be made systematically using the following steps. First, find all possible one atom binding sites on close-packed surfaces (top, bridge, and hollow sites; inset 1 in Fig. 2a). Second, enumerate two-atom configurations by exhaustively evaluating (1) the number of metal atoms that participate in two binding sites, e.g., an atom involved in bonding of two bridge sites, and (2) the total number of the adsorbate-surface bonds–1, 2, and 3 for top, bridge, and hollow sites (Fig. 2a). Third, remove unreasonable configurations of unrealistic bond distances. Fourth, convert the two-atom configurations (e.g., green box in inset 2 in Fig. 2a) to graph transformation rules (e.g., blue box in inset 2 of Fig. 2a). A rule consists of a pattern graph (left-hand side of the blue box) and a replacement graph (right-hand side of the blue box). A graph transformation is applied to a configuration by searching

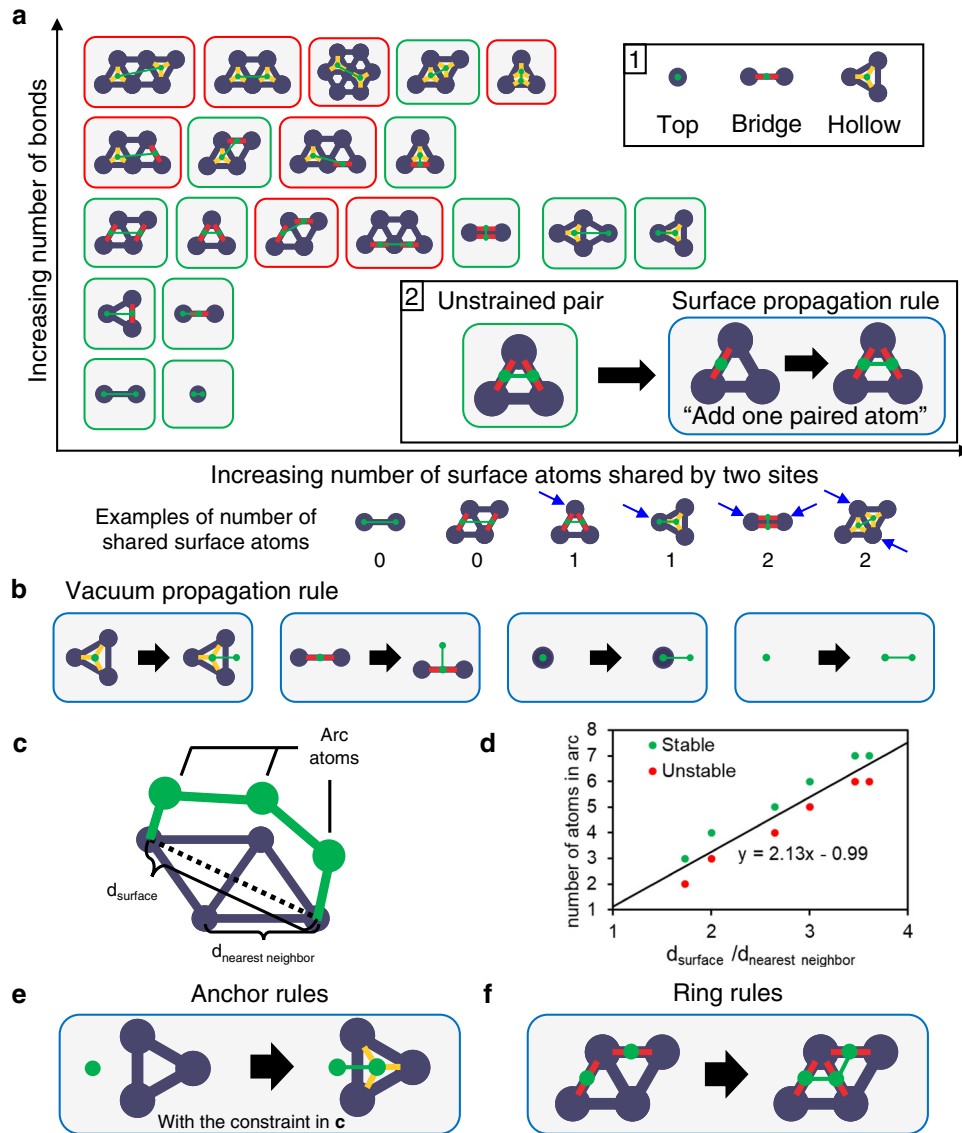

**Fig. 2 Enumeration rules for generating adsorption configurations. a** Surface propagation rules developed by exhaustively assessing two-atom configurations in two dimensions, the number of binding sites sharing one or two metal atoms, and the total number of adsorbate-surface bonds (1, 2, and 3 for top, bridge, and hollow sites, respectively). Configurations in the red boxes are omitted due to their unreasonable geometry. Inset 1: a graphic representation of top, bridge, and hollow sites. Inset 2: graphical representation of the unstrained atom pair configurations converted into a graph transformation rule. **b** Graph transformation rules for the addition of non-surface-bonding atoms. **c** Rules for configurations with an "arc" employ the distances between the anchoring surface atoms $d_{surface}$ and the two nearest neighbor surface metals atoms $d_{nearest\ neighbor}$, respectively. **d** Stability based on the descriptor computed using DFT with the $(CH_2)_x$ adsorbate on Pt(111) as a probe adsorbate. **e** An anchoring rule adds an adsorbed carbon to a non-surface-bonding carbon under the descriptor constraint in **d**. **f** A ring rule constructs a ring by adding an adsorbed carbon to two adjacent carbons. The purple edges and nodes represent the close-packed surface lattice. The red and yellow lines indicate bridge and hollow site bonds with adsorbate atoms, respectively. The green nodes and edges represent the adsorbate atoms and the bonds between them.

for the occurrence of the pattern graph in the configuration, and by replacing the found occurrence with the replacement graph. The two-atom configuration (the green box) becomes the replacement graph (right side of the blue box). The pattern graph (left side of the blue box) is made by removing an atom in two-atom-configurations (the green box). The key postulates are (1) the systematic enumeration of all possible two-atom configurations and (2) the larger configurations consist of two-atom configurations (e.g., a six-atom skeleton can be decomposed to the two-atom configurations). This framework applies to other planar surfaces, such as fcc(100), hcp(1010), and bcc(110).

The second type of rule accounts for non-surface-bonding carbons (e.g., -CH$_2$- and -CH$_3$). Non-surface-bonding carbon can

be added to an adsorbed carbon on top, bridge, or hollow site. Also, non-surface-bonding carbon can be added to another non-surface-bonding carbon to increase the chain length. We call these rules vacuum propagation rules (Fig. 2b).

As shown in Fig. 2c, adsorbates form an "arc" containing a non-surface-bonding atom chain and two anchoring adsorbed atoms (e.g., $(CH_2)_x$). Rules that add an adsorbed carbon to a non-surface-bonding atom can be used to construct arcs, but two anchoring atoms cannot be too far apart. Thus we introduce two metrics, as shown in Fig. 2c, where $d_{surface}$ and $d_{nearest\ neighbor}$ are the distance between the two anchoring surface atoms and the distance between two nearest neighbor surface atoms, respectively. The ratio of the two defines a normalized length threshold

for the arc to be stable (Fig. 2d), which we estimate using DFT with $(CH_2)_x$ on Pt(111). The line between the stable and unstable data indicates the decision boundary we used to decide the arcs' stability. Figure 2e demonstrates a rule for anchoring an arc (called anchoring rules), the pattern graph of which has to respect the distance constraint of Fig. 2d.

The last type of rule adds an adsorbed carbon to two adsorbed carbons, forming a ring (ring rules). Figure 2f shows the ring rules developed by enumerating three adsorbed-carbon chains and building the pattern graph by removing the central atom.

After the enumeration, surface atoms in each enumerated configuration are systematically pruned to build a unique, unambiguous graph (see Supplementary Fig. 1). Duplicate configurations are removed by comparing their hash, such as the SMILES string.

**Force field screening**. We remove strained configurations by optimizing the structures of skeleton configurations with the universal force field[32] with additional interactions between the adsorbate and the surface (see methods for details) with heuristic parameters. The structures with C–C bond lengths outside the range of 0.8 Å and 1.65 Å are removed, which is a broad threshold based on the covalent radius of carbon and oxygen.

**Transformation to an adsorbate**. The unstrained skeleton configurations produce realistic configurations on which we substitute carbon with oxygen at all possible locations and add hydrogens to carbons and oxygens while respecting the valency rule. A varying number of hydrogens is added to the skeleton to represent all possible degrees of saturation; thus, the number of configurations significantly increases in this step.

**Multi-fidelity DFT screening**. We perform low-fidelity DFT calculations of configurations with ≤3 heteroatoms with an early stopping criterion upon configuration divergence to assess the stability. The parameters used for the low-fidelity DFT setup result in less accurate but more efficient calculations (see methods). These achieve decent accuracy compared to the standard DFT relaxation (see methods). The configuration of the DFT-calculated structures is built by determining the connectivity between atoms using $d_{ij} < t(r_{cov,i} + r_{cov,j})$, where $d_{ij}$ is the distance between atoms $i$ and $j$, $t$ is the tolerance factor (1.18 used), and $r_{cov,i}$ is the covalent radius of atom $i$. The stable configurations are further refined using high-fidelity DFT calculations.

**ML-based stability prediction**. We rapidly screen the stability of the configurations with >3 heteroatoms by introducing a fingerprint-like descriptor-based logistic regression (FLDLR), shown in Fig. 3a, with fingerprint-like descriptors as input features[33]. In this method, all possible subgraphs of adsorbate are enumerated, and, for each subgraph, surface atoms connected to the adsorbate are added. The output feature vector contains the number of occurrences for each fingerprint. The training data set is obtained by performing DFT calculations for configurations with ≤3 heteroatoms where the stability is quantified as 1 (stable) or 0 (unstable). A configuration is labeled stable if the connectivity does not change after the DFT relaxation (i.e. the configuration represents a local or global minimum on the potential surface). If the connectivity pattern changes upon DFT relaxation, we labeled them unstable, as the configuration represents an unstable point on the potential surface. As the model will primarily be used to predict configurations of larger adsorbates, we devise a similar extrapolation test. We train the model with adsorbates of ≤2 heteroatoms and assess its error on adsorbates with three heteroatoms. Logistic regression calculates

the probability (a continuous value between zero and one) that a configuration is stable. The probability threshold is used as a tunable parameter for screening. Its effect on the model performance is assessed by the test set recall, precision, $F_1$ score, selectivity, and accuracy in Fig. 3b–e, and Supplementary Fig. 2. As we are interested in a comprehensive database containing nearly all stable configurations, a high recall TP/(TP + FN) value is desired. Here T, F, P, and N are true, false, positive, and negative, respectively. A low threshold of 0.2 (Fig. 3b) ensures that 95% of all stable configurations are sampled (a high recall). However, a low threshold implies also that unstable configurations are also selected (undesired). The precision TP/(TP + FP) in Fig. 3c shows that only 10% of the selected configurations will be stable (a low precision). The $F_1$ score in Fig. 3d shows the harmonic mean of the precision and recall. A threshold of 0.76 most efficiently samples the stable configurations at the cost of unaccounted stable configurations. The selectivity TN/(TN + FP) in Fig. 3e indicates the DFT cost-saving from the ML screening, where we would screen out 44% of the unstable configurations using ML at the threshold of 0.2. Supplementary Figure 2 shows that the accuracy is high at higher tolerance, as most of the enumerated configurations are unstable.

Incorporating FLDLR as a screening tool before performing DFT calculations can significantly reduce the computational cost for larger adsorbates. We retrained the model with ≤3 heteroatoms configurations, and randomly sampled 50 configurations each for 4, 5, and 6 heteroatoms on 11 metals using the uniform distribution over stability score, and performed DFT calculations. The FLDLR calculated score and the DFT inferred stability are compared in Supplementary Fig. 3. We find that 99% of the configurations with low scores (<0.05) are unstable. Since the configurations with low scores (<0.05) comprise most of the large molecule configuration space (84%, 95%, and 99% for 4, 5, and 6 heteroatoms, respectively), one to two orders of magnitude reduction in DFT calculations is expected using the low score as a screening criterion. We believe that ML predictions in the low score region extrapolate well to larger adsorbates; the fingerprints causing instability in the small adsorbate configurations are also present and also cause instability in larger adsorbate configurations. Some of the converged structures with 6 heteroatoms are shown in Fig. 4.

**Enumerated data distribution**. The number of configurations in the various methodological stages is shown in Fig. 5. It increases exponentially with increasing the number of atoms, reaching $\sim 10^8$ configurations for six heteroatoms. The number of DFT-calculated stable structures (green points) scales less steeply than the enumerated ones. The ML screening (using a threshold of 0.2) reduces the number of calculations by two orders of magnitude for adsorbates with 6 heteroatoms.

Figure 6 demonstrates the distribution of the stable configurations assessed using DFT. Intuitively, the atom valency, defined here as (number of the electrons in the valence shell)–(number of neighbor adsorbate atoms), generally follows the number of the coordinated surface atoms (1, 2, and 3 for top, bridge, and hollow sites) as shown in Fig. 6a. Many configurations violate this traditional rule, demonstrating the importance of exhaustive enumeration compared to simple intuition. For complex multi-dentate adsorption, the valency of a single heteroatom is not the only dictating principle; strain effects from stretching the bonds to accommodate the metal lattice and the adsorption characteristics of the other atoms collectively matter. Minimizing the energy of the entire species is the overarching principle. Figure 6b shows the principal component analysis of the stable configurations. The binary matrix is constructed with dimension (number of metals) × (number of configurations), where the matrix element is set to 1 if

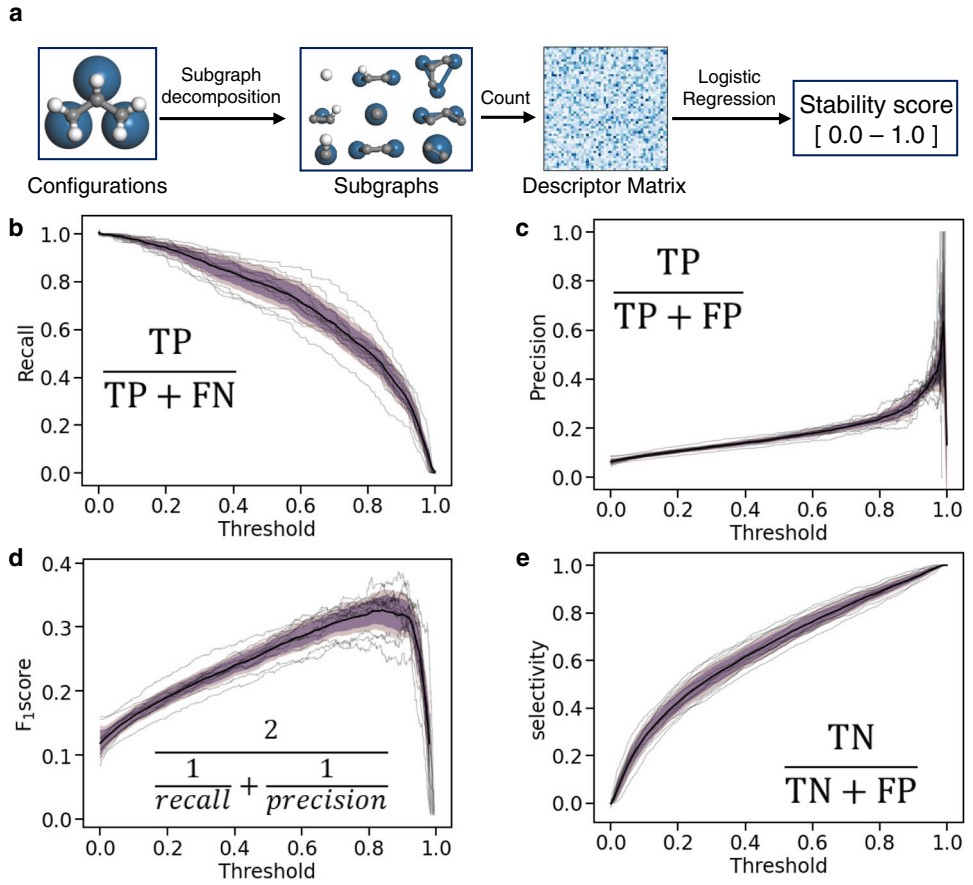

**Fig. 3 Workflow and performance metrics of fingerprint-like descriptor-based logistic regression (FLDLR) of the configuration stability. a** The subgraph space is decomposed using the method described in ref. [33]. **b–e** shows the recall, precision, $F_1$ score, and selectivity of the resulting model for 11 surfaces, respectively. The inset indicates the metric equation where T, F, P, and N indicate true, false, positive, and negative, respectively. The threshold is a tunable probability decision boundary to predict stability, which results in different performances for the metrics considered in this study. The light and dark shaded region indicates 99 and 95% confidence interval, respectively, and the gray and black lines indicate values for 11 surfaces and their mean. The model is trained with adsorbates with ≤2 C and O atoms and tested on adsorbates with 3 C and O atoms.

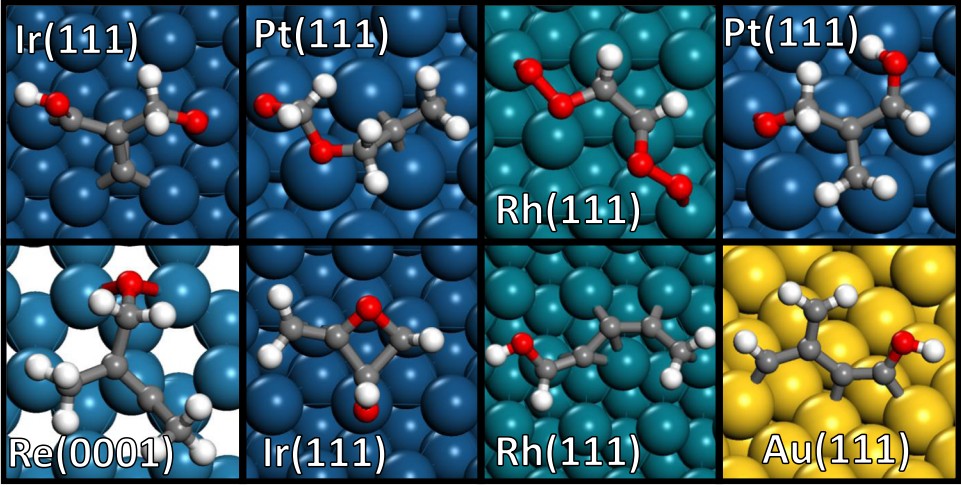

**Fig. 4 Selected stable structures of large adsorbate configurations.** These structures are obtained by performing FLDLR score-based selection, followed by DFT calculations, demonstrating the diversity of the configurations generated by the algorithm.

the given configuration is observed and 0 otherwise (configuration stability matrix in Fig. 6b). We observe that metals form clusters of data. Pt, Pd, Re, Ru, and Ir favor intuitive valency-based configurations: the adsorbate heteroatom valency matches the number of adsorbate-surface bonds (e.g., top, bridge, and hollow sites for $CH_3$, $CH_2$, and $CH$, respectively). Adsorbates fulfill their valency by making the necessary number of bonds with the metal atoms. For Ag and Cu, the number of adsorbate-surface bonds is less than or equal to the valency. The weakly binding Au has the lowest number of stable configurations. Ni and Co contain

structures where the number of adsorbate-surface bonds exceeds the adsorbate atom valency. In this regard, several theoretical and experimental investigations reported that the methyl radical on hollow sites makes three adsorbate-surface bonds (3) and exceeds its valency (1)[34–36] The hollow site adsorption is attributed to the $d$-band coupling with the adsorbate orbitals[34]. This observation is furthermore validated as the adsorption becomes stronger for metal with a $d$-band center closer to the Fermi level. Similarly, the $d$-band center has also been shown to correlate to the energy of

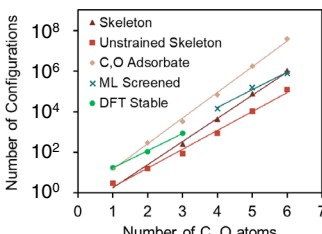

**Fig. 5 Number of configurations at various pipeline stages of Fig. 1b.** All types of configurations scale exponentially with the number of C and O atoms. For configurations with ≤3 heteroatoms, we have explicitly calculated the stability using DFT, whereas, for configurations in larger adsorbates, we used FLDLR[33] with a threshold of 0.2.

adsorbates with a varying number of adsorbate-surface bonds[37]. Thus, we calculated the $d$-band center relative to the Fermi level for the metals considered here: Co > Ni > Rh > Ru > Pd > Re > Cu > Pt > Ir > Au > Ag (Supplementary Table 1). The excess adsorbate-surface bonds for Rh, Ni, and Co are due to their enhanced $d$-band center interaction. Finally, Pt, Pd, and especially Au disfavor η mode interaction between the π-orbit and the metal atoms. As a result, the C=O and C=C substructure is observed less on these metals.

Some molecules do not adsorb on some metals. For example, ethylene ($CH_2CH_2$) does not adsorb on Au(111) and Ag(111) but adsorbs on Cu(111) in η adsorption mode, in agreement with previous DFT calculations[38]. Thus, we perform a principal component analysis of a binary matrix with dimension (number of metals) × (number of molecules), where the matrix element is set to 1 if the given molecule adsorbs on metals and 0 otherwise (molecule adsorption stability matrix in Fig. 6c). Compared to the previous matrix, the second dimension runs over molecules. There are essentially three clusters of data: the first cluster contains mostly strongly binding metals (Pt, Ni, Rh, Co, Ir, Re, Pd, and Ru). On these metals, most of the molecules have multiple stable adsorbed configurations. The second includes several multidentate molecules and molecules with high valency that do not adsorb on Au. This explains the poor performance of Au for C–C scission (encountered, for example, in steam and dry

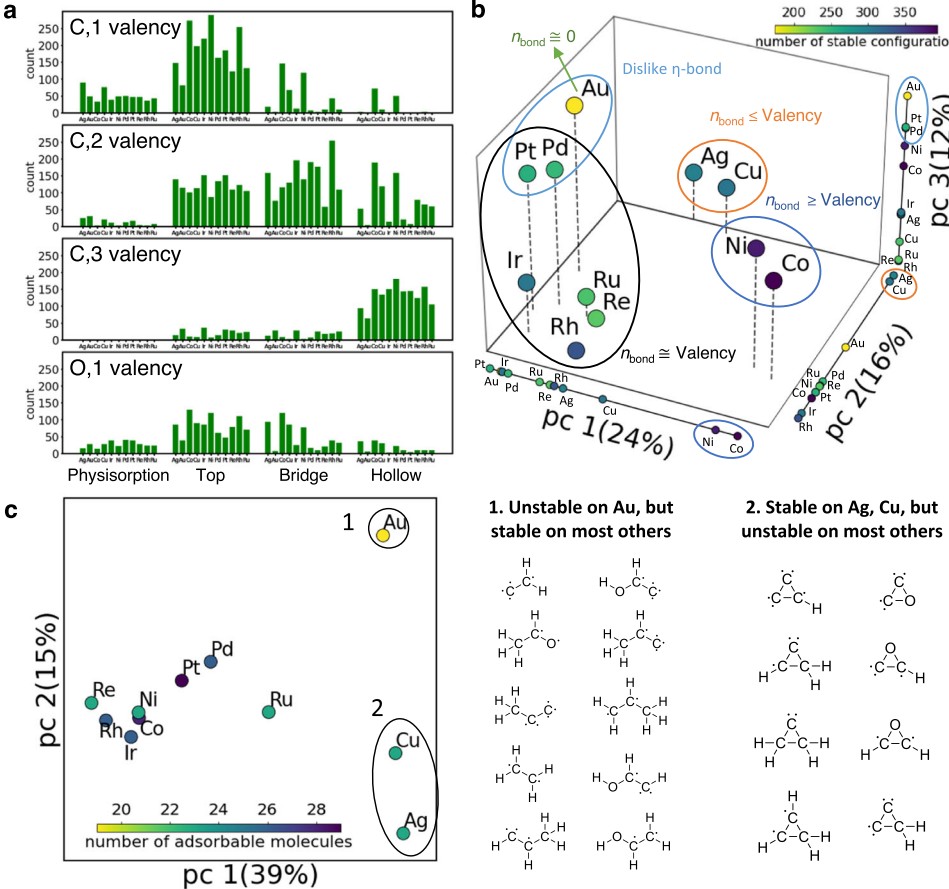

**Fig. 6 Analysis of stable adsorbate configuration data. a** Binding site distribution of stable configurations based on the atoms' element and valency (number of the electron in valence shell)–(number of neighbor adsorbate atoms). Here, C,x valency indicates the carbon atom with x valency. **b** Principal component (pc) analysis of the configurations stability matrix. Four stable configuration groups are discovered based on the number of bonds and the adsorbed atoms' valency. The percentage in the axis indicates the explained variance of the matrix. **c** Principal component analysis of the molecule adsorption stability matrix. Two outlier clusters are observed where (1) several molecules with high valency do not adsorb on Au and (2) some ring molecules are stable on Ag and Cu. The corresponding structures are depicted on the right of the panel.

reforming of larger fuels, e.g., ethanol) and isomerization, as important dehydrogenated reaction intermediates, such as $CH_3CHO$, $CH_2CH_2$, and $CH_2C$, do not adsorb on Au[39,40]. Similarly, Au is a poor catalyst for the Fischer Tropsch synthesis as important intermediates for C–C coupling typically have high valency[41–43]. The third contains Ag and Cu that can adsorb three atom-ring structures that are unstable on other metals which typically dissociate. Some of these molecules are dehydrogenated ethylene oxide (epoxide). Ag and Cu have long been used for selectively producing ethylene oxide[44,45]. Hence, these metals' affinity for the stable ethylene oxide derivatives may be the key to their high selectivity.

**Predicting selective catalysts**. Exploiting the concept of stability of adsorbates being crucial for selectivity, we predict selective catalysts for four heteroatom closed-shell molecules using ethylene oxide as a reactant. We enumerate all possible reaction paths between ethylene oxide and four heteroatom closed-shell molecules by adding and removing C, H, and O in the enumeration rules. For each metal, the shortest reaction paths containing stable intermediates were extracted. The stability of adsorbates was assessed using DFT for ≤3 heteroatom adsorbates and FLDLR with a threshold of 0.95 (high probability of stability) for >3 heteroatom adsorbates. The paths to closed-shell molecules with less than 5 viable metals are shown in Fig. 7 as examples of selective catalysts. The thermochemistry and kinetics were not assessed, thus realizing these chemistries requires further investigation. We find that Au(111) is selective to all nine molecules whereas seven other surfaces are selective to a few. Especially, eight out of nine molecules contain rings, which are typically produced by homogeneous organic reactions. Specifically, homogeneous gold catalysts produce small rings with less than six atoms[46], and some cyclization transfers to gold nanoparticles[47]. These facts indicate that the discovered pathways could be experimentally viable.

## Discussion

The conversion of large molecules is poorly understood due to the large size of the reaction network and the lack of automation for initializing DFT calculations of large adsorbates. This, in turn, stems

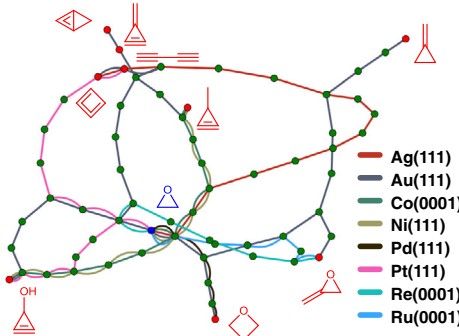

**Fig. 7 Pathways from ethylene oxide to metal-selective closed-shell molecules.** Pathways consist of adsorbates and molecules represented by nodes, which are connected by lines, representing elementary reactions. An elementary reaction can add or remove C, H, and O, and perform C–C coupling. Blue, green, and red circles indicate the reactant ethylene oxide, intermediates, and closed-shelled molecules, respectively. The color of the lines indicates the metal surface that has stable intermediates between ethylene oxide and close-shelled molecules. Closed-shell molecules with less than five metals are shown. The stability of ≤3 heteroatom adsorbates are assessed using DFT and >3 heteroatom adsorbates using FLDLR with a threshold of 0.95.

from the combinatorial explosion of complex adsorbate configurations that dictate thermochemistry and reaction pathways. The intuitive binding of adsorbates, based on the heteroatom valency, has long been used. We discover it can fail, yet certain clusters of data are observed based on the $d$-band/adsorbate orbital interaction. To the best of our knowledge, this work presents the first systematic enumeration of multidentate adsorbate configurations with arbitrary binding motifs. Importantly, we also find correlations between configurations with the $d$-band. We observe that the stability of intermediates is essential for highly selective catalysis, as a correlation between the intermediate stability and selectivity is demonstrated for the ethylene oxide and Fischer Tropsch process. More generally, a catalyst cannot produce a molecule if its reaction intermediates are not stable on it, and the library of molecules we built can be leveraged to understand if a metal catalyst can conduct specific chemistry. Potentially, highly selective catalysts can be made by designing catalytic sites that selectively adsorb desired adsorbates. Furthermore, we often assume in creating volcano curves for materials discovery that the reaction pathway, intermediates, and rate-determining step are the same on all catalysts. Our results clearly identify clusters of materials for which this is true but expose profound differences among clusters. The developed database could aid in the theoretical investigation of large molecules by predicting adsorbate thermodynamic properties and enabling a database for lateral interaction models[30], Brønsted–Evans–Polanyi relations[48] (scaling relationship between reaction energy and activation energy), and transition state structures. These investigations could enable microkinetic model development toward elucidating catalyst design principles. We emphasize that, while we focused on the widely studied close-packed surfaces, the framework can be expanded to other surfaces such as fcc(100), stepped surfaces, and alloys by constructing an appropriate surface lattice, and differentiating surface atoms by elements and location (e.g., step-edge, corner, terrace). Other heteroatoms, such as nitrogen and sulfur, with pharmaceutical applications, can trivially be considered.

The number of enumerated configurations becomes computationally vast, reaching $10^8$ for adsorbates with six C and O atoms, posing a significant challenge in studying large molecules. The difference in the slopes of enumerated configurations and DFT-calculated stable configurations is notable, underscoring that an improved enumeration algorithm could potentially be developed. We expect the performance of the ML model to improve significantly by adding structures of four C and O atoms, as an adsorbed carbon has a maximum of three neighbors. We are expanding the database to improve the ML model.

Our scheme can be further improved in several directions. Lateral interactions between adsorbates are well-known to affect the adsorption energy and potentially change the preferred site[49]. While we used a relatively low coverage, the effect of lateral interactions on the configuration stability remains unclear. We also did not assess the vibrational modes of adsorbates, and thus, some adsorbates may be on unstable saddle points on the potential energy surface. Our scheme faces an additional challenge for larger biomass molecules, such as glucose involving 12 C, O atoms, requiring $>10^6$ DFT calculations. Potentially online learning, where we repeat the cycle of data sampling and model training, can improve model accuracy and reduce the number of candidates continuously on the fly. Our scheme has similarities with global optimization techniques aiming to identify all minima in a high-dimensional space. Integration with advanced global optimization algorithms[50–52] can improve scalability as well. As we focused on the enumeration of adsorbates' connectivity patterns, our scheme does not account for cis/trans isomers not implicitly accounted for by the connectivity pattern (Supplementary Fig. 4). The assessment of the quality of the data is critical. While we addressed the challenge of the enumeration of

**Table 1 Added distance constraint for the force field optimization.**

| Distance constraint | $k$ (kcal mol$^{-1}$ Å$^{-2}$) | $r_{eq}$ (Å) |
|---|---|---|
| SA-its lattice position, plane direction | $10^5$ | 0 |
| SA-its lattice position, normal direction | $10^3$ | 0 |
| AA-SA | $4 \times 10^3$ | 2 |

SA and AA indicate surface atom and adsorbate atom, respectively.

**Table 2 Added angle constraint for the force field optimization.**

| Angle constraint | $k$ (kcal mol$^{-1}$ rad$^{-2}$) | $\theta_{eq}$ (°) |
|---|---|---|
| SA-carbon (1 valency)-AA | 300 | 109.5 |
| SA-carbon (2 valency)-AA | 150 | 145 |
| SA-carbon (3 valency)-AA | 150 | 180 |
| SA-oxygen-AA | 150 | 120 |
| AA-AA (1 valency)-AA | 200 | 109.5 |
| AA-AA (2 valency)-AA | 200 | 145 |
| AA-AA (3 valency)-AA | 200 | 180 |
| SA-AA-SA | 4000 | 109.5 |

SA and AA indicate surface atom and adsorbate atom, respectively.

connectivity patterns, future work should include the curation of the data, which can include manual curation, and the use of statistics to identify faulty data.

## Methods

**Force field optimization**. The universal force field as implemented in RdKit (Rdkit.org) is modified to generate the structures. In addition to the standard UFF parameters, distance and angle constraints are added using the quadratic relations,

$$E_r = 1/2 k(r - r_{eq})^2$$
$$E_\theta = k(\theta - \theta_{eq})^2 \qquad (1)$$

where $E_r$ and $E_\theta$ are the distance and angle energy, $k$ is the force constant, $r$ and $\theta$ are radius and angle, and the subscript eq represents the equilibrium value. Forces that hold surface atoms in their lattice position and describe adsorbate atom–surface atom bond are added, as shown in Table 1. Also, various angle constraining forces are added to generate reasonable structures, as shown in Table 2. The heuristic forces provide a plausible initial guess structure for DFT calculations, typically better than the manually guessed structures. As a strong force constant is used for the adsorbate-surface bond, the strain manifests as the distorted adsorbate-adsorbate bond, which we used to decide the strained, unstable configurations.

**DFT calculations**. We performed DFT calculations using the Vienna ab initio Simulation Package[53]. The electron exchange and correlation energies were computed using the PBE functional[54]. Our previous study finds that the choice of functional and dispersion correction does not affect the geometry of a large molecule, namely furan, significantly[55]. The core electrons were calculated with the projector augmented-wave (PAW) pseudopotentials[56]. The Brillouin zone is sampled with a Methfessel-Paxton smearing of 0.1 eV[57].

To construct the slab, the lattice constants of the metals are optimized using $15 \times 15 \times 15$ Monkhorst-Pack $k$-point mesh with Blöchl correction[58,59], D3 dispersion correction[60], and the plane-wave cutoff energy of 500 eV. Close-packed surfaces (fcc(111), hcp(0001)) were modeled with a four-layer deep $4 \times 4$ unit cell with a 20 Å vacuum where the bottom two layers are fixed.

For assessing configuration stability, we used low-fidelity parameters. The cutoff energy of 300 eV was used with non-spin polarized calculations. Gamma point was used to sample the Brillouin zone. The quasi–Newton algorithm was used to converge the structure into its instantaneous ground state. The DFT calculations were stopped if the configuration diverged to another configuration. Molecular graphs are constructed by adding an edge between two atoms if the distance between the two is less than the sum of the two elements' covalent radius multiplied by 1.18. If the calculation did not converge after 200–800 ionic steps, we used the conjugate-gradient algorithm to relax the structure. Here, the early stopping is not used to observe the final structure. The configuration graph is determined using covalent radius-based graph construction[33]. To test the stability convergence, we compare the stability of adsorbates with ≤2 C, and O atoms on

Pd(111) between the low-fidelity and standard DFT parameters. The high-fidelity calculations entail cutoff energy of 400 eV with $3 \times 3 \times 1$ Monkhorst-Pack $k$-point mesh[58], spin-polarization, and D3 dispersion correction[60]. Here, 9 out of the 52 stable configurations of standard DFT calculations diverged in low-fidelity (see confusion matrix in Supplementary Table 2). Out of these, the binding energies of the four configurations are >0.5 eV higher than the ground state configuration binding energy of the respective adsorbate. Four configurations are local ground states (0.13, 0.19, 0.12, and 0.01 eV with respect to each molecules' ground state configuration). Only one ground state configuration was not predicted stable in the low-fidelity calculation. This was due to the early stopping method, stopping calculations prematurely before convergence.

## Data availability
The enumerated configurations, their stability, and energetics are available at our GitHub repository[61].

## Code availability
The enumeration and machine learning code with an example output is available at our GitHub repository[61].

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

## Acknowledgements

This work was supported by the National Research Foundation of Korea, the Ministry of Science and ICT under award numbers 2021R1C1C2094407 (G.G.) and 2019M3D3A1A01069099 (Y.J.), and as part of the Catalysis Center for Energy Innovation, an Energy Frontier Research Center funded by the US Department of Energy, Office of Science, Office of Basic Energy Sciences under award number DE-SC0001004 (D.G.V. and M.L). We acknowledge the Korea Institute of Science and Technology Information (KISTI) for the computational resources provided for this research.

## Author contributions

G.G. conceived this project and developed the enumeration algorithm, and the ML model, and analyzed the configuration space. G.G. and M.L. performed DFT calculations. G.G., Y.J., and D.G.V. discussed the results and assisted with the manuscript preparation.

## Competing interests

The authors declare no competing interests.
