## [Peer Review File · Nature Communications]

REVIEWER COMMENTS

Reviewer #1 (Remarks to the Author):

The paper "Automated Exploitation of the Big Configuration Space of Large Adsorbates on Transition Metals Reveals Chemistry Feasibility" addresses the challenging and very important problem of evaluating stability of various adsorption configurations for large molecules. The authors develop a methodology to address this problem, based on their earlier work. While the topic is certainly of great interest to the broad readership of Nature Communications, the manuscript requires a revision because of somewhat convoluted writing and missing important information. My detailed comments are given below.

Major comments:

1) The machine-learning part description is convoluted. It is unclear which exactly properties are evaluated (how the stability is quantified), and what is the accuracy of the prediction. Figure 5 contains some information on the model's accuracy, but the description of the graphs is very unclear. This should be improved.

2) All assumptions should be clearly stated in the manuscript. For example, is it assumed that adsorbate-adsorbate interactions are small? It is clearly assumed that vibrational contributions do not affect stability. Also, the choice of the particular DFT functional should be explained.

3) While some examples and analysis are given, there is no new prediction based on the novel methodology (e.g., for molecules with $>3C,O$) that could be validated by experiments. It would be exciting and important to add such a prediction.

Minor comments:

"As a configuration is built, occupied surface atoms are removed from the list of unoccupied surface atoms to ensure uniqueness and eliminate ambiguity (see Figure S1)." - the meaning of "occupied/unoccupied surface atoms" is unclear, and does not read well.

"We observe that the stability of intermediates is essential for highly selectivity catalysis, as the ethylene oxide and Fischer Tropsch cases reveal." - use of "cases" here is cryptic; please extend and improve wording

"More generally, a catalyst cannot produce a molecule whose intermediates are not stable on it" - "molecule whose intermediates" is not a good wording

"BEP relations" - BEP abbreviation should be introduced

"We emphasize that, while we focused on the widely studied closely packed surfaces, the framework can be expanded to other surfaces." - this should be discussed a bit more; how the methodology can be extended to stepped surfaces, alloys? will it still be feasible?

It is unclear if a dispersion correction was used in adsorption energy calculations, this should be clarified. Dispersion is important for large-molecule adsorption.

Was spin considered? This should be clarified and the choice should be explained.

It would be useful to show some structures of the larger molecules calculated by DFT.

Reviewer #2 (Remarks to the Author):

I am reviewing the paper of Gu, Lee, Jung, and Vlachos. They present a systematic methodology to explore the configuration space of larger adsorbates, demonstrated with molecules with up to 6 atoms in their backbone (C/O) and go well beyond two unsaturations. The process starts in a rather unconventional way, (1) first building the bare backbone made only of C atoms, without hydrogen atoms. They are then (2) pruned out of possible unphysically strained structures. Then (3) some C atoms are replaced by O. Finally, (4a) large molecules are screened via a ML approach whereas (4b) smaller molecules are screened with low- and high-quality DFT runs. This way, the authors provide a tool to solve an important problem, that is the systematic exploration of the conformational space of large molecules, such as those contained in biomass (There are however alternatives that still work, without being that extensive and systematic). Further analyses were done via a PCA to provide a rough explanation about how the chemical space looks like.

I have found some results quite impressive and unexpected. For instance, the fact that they described hypervalent binding modes. For instance, CH₃ has been experimentally found to bind to threefold sites on Cu(111) and perhaps other metals. Thus, the C there is strangely coordinated with 6 atoms. This is a well-established result, that also agrees with theoretical simulations. Yet, all other systematic exploration methods I have found out there have been unable to reproduce this.

Besides, the authors walked an extra mile when they rationalized their results by helping themselves with a PCA. The chemical space of the metals under study forms clusters that can be related to the number of bonds between the adsorbates and the surface.

I have found the methodology rather robust. As Nature Communications focuses on a broad set of readers, the authors need to walk yet another extra mile to assure it is understandable for such an audience. Most of my comments go in this direction. Besides, the authors made available their results, but not the code used to reach such results. I think the Manuscript would only fit Nature Communications tight standards if the code is made fully available (and submitted to peer review in the next round) and a few use examples are included. Otherwise, the paper would better fit Scientific Reports.

Regarding the "hypervalent" adsorption states. The authors may extend this discussion and, especially, cite experimental studies. They need to showcase this is a well-established result rather than a weirdness of DFT simulations (that is what my coworkers and I thought the first time we saw that, some years ago). Currently, only a theoretical study is cited.

The method should also work for molecules containing N, S, or other heteroatoms. Such molecules are important too in the pharmaceutical industry. The authors may mention it.

There are features that make this paper unnecessary complex to understand unless it is re-read several times. For instance, the procedure (which is already complex) is not stated systematically. It is rather outlined three times providing complementary information in lines 62-77, 79-114, 115-147. Such paragraphs also contain many lateral notes, dangling references, and redundancies that drives the reader's mind out of the context, such as:

* "see below" (all instances)

* line 102 "modified force field" (which modifications? it appears just a common quadratic FF).

* Line 146 "enumerating $\times 2$ "

In line with the previous comment, each "rule" (or step?) needs to be fully and clearly explained before moving to the next one (ie, merge lines 79-114 and 115-147 into a single story, each paragraph fully explaining one step). The authors mention "two types of rules" out of context in line 137 that are then never more explained. Are they rules by themselves? When is the recursion of lines 132-136 applied?

Also in line with the previous comments, the procedure in Figure 1b needs to be integrated better with the text and Figure 1a. For instance, it is not clear when high-quality DFT is applied.

I have read 7+ times lines 125-129 and still do not understand rule 4. Did the authors mean that they reduced their set to structural graphs that are equivalent by symmetry? The full paragraph needs to be rewritten.

How did the authors check that the result of the relaxation was OK? Did they check a connectivity matrix, for instance? To be more specific:

* There are structures very relevant for catalysis that are metastable structures (eg, OCCO). When one relaxes such structures, they may either break (eg, to form CO+CO), or tautomerize. Tautomerization is exponentially common on "large" molecules, containing at least 4 C+O atoms. Also, rearrangements may fully change the backbone, thus transforming a "linear" initial guess into a "cyclic" molecule, or even fully moving a functional group to another atom. So, how can they assess that two particular conformations do belong to the same molecule?

* The authors need to comment on how many of such skeletons were branched or cyclic. Also, how did they assess cycles of three atoms? From Equation 1 and Table 2, it appears that 3-atoms rings (angles around 60°) would all be deemed unstable.

* How did they deal with cis-trans isomerism?

* Most importantly, if the input data for the logistic regression is not even partially trained with large molecules (let us say, 5% or 10% of them), how can they describe faithfully the aforementioned complexity sources, especially the tautomerizations?

There are many apparent misnomers used throughout the manuscript:

* The very word "rules". "steps" seems to fit the context much better.

* "enumeration", in "graph enumeration rules". Perhaps "amplification" or "systematic construction" would be more descriptive.

* "Machine Learning"/"ML". As a general criticism to our field: "ML" is becoming a wildcard/generic word, often being used to avoid explaining relevant methodological details or even hiding profound inconsistencies. The authors need to escape such a trap and be crystal clear about their true "ML" core. The fingerprint-like descriptor and how they coupled it with the logistic regression must be fully and dully explained in its corresponding context, even if part of it is fully explained in a previous study. (Are the "fingerprints" equivalent to the "rules" or where are they?). They need a name for the method (eg FDLR, from "fingerprint-like ...", line 70?) and mention that with a clear, consistent label every time they did use it.

* Related to the previous point, PCA is sometimes used as ML, but here the authors used it to get insights, but not as part of their main method. Thus, replacing all instances of "ML" with the name of the specific method used in each step is a must.

* "Heteroatom". The meaning of "heteroatom" in our specialized field (an atom that is not hydrogen, mostly used in theoretical heterogeneous catalysis) differs from that of IUPAC (an atom that is not C or H, mostly used in organic chemistry). By the context, I am not fully sure which one the authors wanted to convey. The authors would better state the intended meaning in the first use (introduction, line 47).

* "Multi-fidelity DFT". The meaning of this becomes clear once one has read 60% of the manuscript, but not the first times it appears.

* "Manifold". A "topological space that locally resembles Euclidean space near each point" does not seem to fit in the contexts such word was used. Do the authors mean "density" of the surface?

* "Valency" (valence), especially in the context of Figure 5, it is more of a "coordination" or "maximum coordination" or "degree of unsaturation". See again IUPAC gold book.

Minor clarifications/typos:

* line 32: activation energies are not thermodynamic parameters, but kinetic parameters.

* line 33: Adsorbate configurations are "prerequisites to compute activation energies". It may read better than "initial or final states in computing activation energies"

* line 43: In this Reviewer's opinion, the challenge is not only the computational cost but the quality of the data obtained when automating. As the volume of results is too high, if the results are not curated by a human, there will always be hidden major problems, not envisaged by the ones doing the automation.

* lines 63 and 320, to be replaced by "closed-paced".

* line 134: Point 2 unclear: Is the 6-atom backbone reduced to pairs of atoms?

* line 150-152: unclear sentence. Did it mean "As a configuration is built, occupied surface atoms are not considered for the next iteration"?

* Figure 2a, inset 3, is very unclear. The text explaining it is also unclear.

* Figure 2c: What is the physical meaning of the regression parameters 2.13 and -0.99? It is deeply unclear how it can be related to stability.

* line 175: please define "hyperparameter" for a broad audience.

* line 183: why the harmonic mean? why not the geometric mean, or even a fully new descriptor with a clear interpretation such as $(TP+TN)/(TP+TN+FP+FN)$.

* Figure 3 b-e: The terms in the "y" axis are very hard to understand. A clear definition (equation) of each quantity may be provided as insets. Besides, the methods used to get the 95% and 99% must be fully explained.

* line 228: The number of adsorbate-surface bonds and the d-band centre have been previously linked via a PCA, Nat. Commun 2019, 10, 4687, with rather similar conclusions.

* line 233-251 and Figure 5c: It is not fully clear what is the difference between both PCA. The authors label them "adsorbable molecules" and mention ethylene as an example. Did they mean "closed-shell molecules"?

* line 244: I understand that closed-shell molecules do not adsorb on too noble surfaces such as Au. Yet, I wonder if the CH₂C is a typo (meaning HCCH), or if the authors consider it an "adsorbable molecule" despite being a (probably unstable) biradical in gas-phase.

* Figure 5b: The "intuitive" word is tricky in this context ($n_{\text{bonds}} \approx \text{valence/coordination}$) and I suggest the authors remove it. For instance, my intuition says that acetaldehyde would "physisorb" ($0 = n_{\text{bonds}} < \text{valence} = 2$) rather than "chemisorb" with C-surface and O-surface bonds, for maybe all metals ($n_{\text{bonds}} = \text{valence} = 2$).

* Related to the previous one, how is that formic acid is "unstable" on Au?

* line 277: the word "mechanism" in this context should instead be "path". In a complex reaction network, a "mechanism" is a collection of overlapping "paths". It is true that in old-school mechanistic studies, due to the lack of computer power, only a particular path was assumed for all materials being screened. However, this is seldom done in serious studies nowadays.

* line 282: The authors need to explain what a BEP relation is for a broad audience. I would rather avoid that and would mention how they would translate this method to find transition states (which is 100× more interesting than a BEP relationship!).

* line 324, when doing the low-fi DFT at 300 eV, which pseudopotentials did the authors use for C and O? the "normal" ones, or the C_s O_s version?

* line 339: To which molecule are these "local ground states" related?

Reviewer #3 (Remarks to the Author):

The manuscript presents the results of a configurational search scheme for organics on metal surfaces employing graph representations of configurations, force field optimization, DFT simulations and machine learning. The authors claim to be able to identify the optimal binding configuration of adsorbates of up to 6 C/O atoms. The strength of this manuscript are:

A, This is a significant improvement over similar attempts particularly, ref 27 which can search very limited configurational spaces, and ref 29 and 30 which are similar in spirit but are case specific as opposed to general.

B, The presentation of the algorithm is, for the most part, very clear and well documented.

C, Its potential use in identifying intermediates along a catalytic pathway will be highly valuable to the catalyst community at large.

Nonetheless I have some difficulties with this work.

A, It seems to have very unmeritable scaling to the point where systems with 10 atoms would require 10⁵-10⁶ DFT simulations which is impractical. Yet many many biomass derived molecules will be in this size range or larger. For a first iteration of an algorithm this limitation is to be expected but the authors should admit this limitation and at least discuss potential algorithmic advances to improve upon this to reach large scaling.

.

B, Along these lines the authors have avoided references to global optimization techniques-which is in a sense what they are proposing. A few notable algorithms even claim to be able to scale to large dimensional systems using adaptive learning

Ex

1, Zhang, et al (2020). NWPEsSe: an adaptive-learning global optimization algorithm for nanosized cluster systems. *Journal of chemical theory and computation*, 16(6), 3947-3958.

2, Janet, et al (2020). Accurate multiobjective design in a space of millions of transition metal complexes with neural-network-driven efficient global optimization. *ACS central science*, 6(4), 513-524.

3, Bisbo, M. K., et al. (2020). Efficient global structure optimization with a machine-learned surrogate model. *Physical review letters*, 124(8), 086102.

Approaches such as these may be beneficial to hybridize into the current model to further reduce the number for DFT simulations.

C, The utility and strength of the approach would be best served by presenting an illustrative example of catalyst discovery-ex epoxidation reactions on Ag etc. As it is now, its just a complex configuration space sampling tool and, although interesting, potentially to specialized for nature Comm. However, I do believe the authors can rectify this.

Point-by-point responses and new text in blue.

Reviewer 1

Comment 1: The paper "Automated Exploitation of the Big Configuration Space of Large Adsorbates on Transition Metals Reveals Chemistry Feasibility" addresses the challenging and very important problem of evaluating stability of various adsorption configurations for large molecules. The authors develop a methodology to address this problem, based on their earlier work. While the topic is certainly of great interest to the broad readership of Nature Communications, the manuscript requires a revision because of somewhat convoluted writing and missing important information. My detailed comments are given below.

Response: We are grateful to the reviewer for finding the paper of great interest to the board Nat Comm readership and appreciating the challenge and importance of this problem and for providing insightful comments to improve the manuscript.

Comment 2: The machine-learning part description is convoluted. It is unclear which exactly properties are evaluated (how the stability is quantified), and what is the accuracy of the prediction. Figure 5 contains some information on the model's accuracy, but the description of the graphs is very unclear. This should be improved.

Response: We agree that the model description is not clear. We added "the stability is quantified as 1 (stable) or 0 (unstable)" on page 7 to clarify the stability quantification. We added Figure S2 showing the accuracy of our model and added related discussion, "**Error! Reference source not found.** shows that the accuracy is high at higher tolerance, as most of the enumerated configurations are unstable." on page 8. Regarding the graph description, we believe that the reviewer is pointing to Figure 4 (now Figure 5) instead of Figure 5. We added more description to the caption, namely "All types of configurations scale exponentially with the number of C and O atoms. For the configurations with ≤ 3 heteroatoms, we have explicitly calculated the stability using DFT, whereas for larger adsorbates, we used FLDLR³³ with a threshold of 0.2."

Comment 3: All assumptions should be clearly stated in the manuscript. For example, is it assumed that adsorbate-adsorbate interactions are small? It is clearly assumed that vibrational contributions do not affect stability. Also, the choice of the particular DFT functional should be explained.

Response: We agree with the reviewer. To address this, we added a paragraph discussing the assumptions and limitations of our model at the end of the discussion section, which is "Our scheme can be further improved in several directions. Lateral interactions between adsorbates are well-known to affect the adsorption energy and potentially change the preferred site.⁴⁷ While we used relatively low coverage, the effect of the lateral interaction on the configuration stability remains unclear. We also did not assess the vibrational modes of adsorbates, thus some adsorbate structures may have imaginary frequency. Our scheme faces an additional challenge for larger biomass molecules such as glucose involving 12 C, O atoms, as it requires $>10^6$ DFT calculations.

Online learning can be implemented where we repeat the cycle of data sampling and model training. In this way, the model accuracy improves and the number of candidates reduces continuously on the fly. “Our scheme shares similarities with global optimization techniques which the aim to identify all local and global minima in a high-dimensional space. Integration with these advanced global optimization algorithms⁴⁸⁻⁵⁰ is expected to improve scalability.” As we focused on the enumeration of adsorbates’ connectivity pattern, our scheme does not account for the cis, trans isomerism that is not implicitly accounted for by the connectivity pattern (Figure S4).”

Previously, we compared the effect of functional choice (RPBE vs PBE, and the effect of D3) for a large molecule, furan (*ACS Catal.* **2012**, 2, 2496). We found that the bond distances changed within ± 0.02 Å among functionals and were similar to experimentally determined values. Thus, we expect the particular choice would not likely affect the geometry. We added “Our previous study finds that the choice of functional and dispersion correction does not affect the geometry of a large molecule, furan, significantly.⁵³” on the DFT calculation section in the methods.

Comment 4: While some examples and analysis are given, there is no new prediction based on the novel methodology (e.g., for molecules with $>3C,O$) that could be validated by experiments. It would be exciting and important to add such a prediction.

Response: We thank the reviewer for this great suggestion. We performed multi-fidelity DFT calculations for 1,650 configurations and found that the machine learning model, named FLDLR, can be used as a screening tool to reduce DFT computational cost. While several configurations predicted as stable were not stable using DFT calculations, the algorithm predicts the unstable configurations well. This is very important because most of the configurations with larger molecules have a low score. Thus, we expanded the application of the machine learning model as a screening tool before performing DFT calculations. Figure S3 has been added which compares the predicted stability score and the DFT calculated stability. The following paragraph was added in the machine learning model section:

“Incorporating FLDLR as a screening tool before performing DFT calculations can significantly reduce the computational cost for larger adsorbates. We retrained the model with ≤ 3 heteroatoms configurations, and randomly sampled 50 configurations each for 4, 5, and 6 heteroatoms for 11 metals using the uniform distribution over stability score, and performed DFT calculations. The comparison between the FLDLR calculated score and the DFT calculated stability is shown in Figure S3. We find that 99% of the configurations with low scores (<0.05) are unstable. Since the configurations with low scores (<0.05) comprise most of the large molecule configuration space (84%, 95%, and 99% for 4, 5, and 6 heteroatoms, respectively), one to two orders of magnitude reduction in DFT calculations can be expected using the low score as the screening criterion. We believe that machine learning predictions in the low score region extrapolate well to the larger adsorbates as the fingerprints that cause instability in the small adsorbate configurations also cause instability for larger adsorbate configurations.”

Regarding predicting new catalysts for larger molecules, we have suggested selective catalysts in response also to comment 6 of Reviewer 3. Specifically, we looked for closed-shell molecules with 4 heteroatoms that only a few metals have reaction paths from ethylene oxide. The resulting

network of paths is shown in Figure 7. We find that 9 molecules in this analysis have paths on Au(111), whereas other catalysts are selective to a few. Interestingly, 8 out of 9 closed-shell molecules are ring molecules, which is an exciting prediction as the synthesis of ring compounds typically involve homogeneous reactions.

We added related discussion on pages 12-13. "Predicting selective catalysts. Based on the idea that the stability of the adsorbate is crucial for selectivity, we predict selective catalysts for 4 heteroatom closed-shell molecules using ethylene oxide as a reactant. We enumerated all possible reaction paths between ethylene oxide and 4 heteroatom closed-shell molecules using the addition and removal reaction of C, H, and O as enumeration rules. For each metal, the shortest reaction paths that contain only the stable intermediate adsorbates were extracted. The stability of adsorbates was assessed using DFT for ≤ 3 heteroatom adsorbates, and using FLDLR with a threshold of 0.95 (high probability of stability) for >3 heteroatom adsorbates. The paths to closed-shell molecules with less than 5 viable metals are shown in Figure 7 as examples of selective catalysts. We note that thermochemistry and kinetics are not assessed, thus realizing these chemistries requires further investigation. We find that Au(111) is selective to all 9 molecules in our analysis whereas 7 other surfaces are selective to a few. Especially, 8 out of 9 molecules are ring molecules, which is valuable as the synthesis of ring compounds is typically limited to homogeneous organic reactions. Homogeneous gold catalysts are known to produce small rings (ring with <6 atoms),⁴⁶ and some of the cyclization activities of homogeneous gold catalysts are shown to transfer to gold nanoparticles as well.⁴⁷ These indicate that the discovered pathways could be experimentally viable."

Comment 5: "As a configuration is built, occupied surface atoms are removed from the list of unoccupied surface atoms to ensure uniqueness and eliminate ambiguity (see Figure S1)." - the meaning of "occupied/unoccupied surface atoms" is unclear, and does not read well.

Response: We agree with the reviewer. We changed the sentence to "After the enumeration, surface atoms in each enumerated configuration are systematically pruned to build a unique, unambiguous graph (see Figure S1)." We moved the detailed discussion to the Figure S1 caption and clarified the algorithm with a more descriptive name for the occupied/unoccupied surface atoms.

Comment 6: "We observe that the stability of intermediates is essential for highly selectivity catalysis, as the ethylene oxide and Fischer Tropsch cases reveal." - use of "cases" here is cryptic; please extend and improve wording

Response: We agree with the reviewer, and changed the sentence to "We observe that the stability of intermediates is essential for highly selective catalysis, as the correlation between the intermediate stability and selectivity is demonstrated for the ethylene oxide and Fischer Tropsch process."

Comment 7: "More generally, a catalyst cannot produce a molecule whose intermediates are not stable on it" - "molecule whose intermediates" is not a good wording

Response: We agree with the reviewer, and changed the sentence to "More generally, a catalyst cannot produce a molecule if its reaction intermediates are not stable."

Comment 8: "BEP relations" - BEP abbreviation should be introduced

Response: We thank the reviewer for pointing this out, and the abbreviation is expanded to its full name.

Comment 9: "We emphasize that, while we focused on the widely studied closely packed surfaces, the framework can be expanded to other surfaces." - this should be discussed a bit more; how the methodology can be extended to stepped surfaces, alloys? will it still be feasible?

Response: We agree with the reviewer and expanded this discussion by editing the sentence to "We emphasize that, while we focused on the widely studied closely packed surfaces, the framework can be expanded to other surfaces, such as fcc(100), stepped surfaces and alloys by constructing an appropriate surface lattice, and differentiating surface atoms by elements and location (e.g., step-edge, corner, terrace)."

Comment 10: It is unclear if a dispersion correction was used in adsorption energy calculations, this should be clarified. Dispersion is important for large-molecule adsorption.

Was spin considered? This should be clarified and the choice should be explained.

Response: This is a good point. We performed spin-polarized calculations with the dispersion effect (DFT-D3) in the high-fidelity calculations. We edited the method section to clarify, namely "The high fidelity calculations entail a cutoff energy of 400 eV with $3 \times 3 \times 1$ Monkhorst-Pack k-point mesh,⁵⁶ spin-polarization and the inclusion of D3 dispersion correction.⁵⁸"

Comment 11: It would be useful to show some structures of the larger molecules calculated by DFT.

Response: We agree with the reviewer and added some structures of larger molecules in Figure 4.

Reviewer 2

Comment 1: I am reviewing the paper of Gu, Lee, Jung, and Vlachos. They present a systematic methodology to explore the configuration space of larger adsorbates, demonstrated with molecules with up to 6 atoms in their backbone (C/O) and go well beyond two unsaturations. The process starts in a rather unconventional way, (1) first building the bare backbone made only of C atoms, without hydrogen atoms. They are then (2) pruned out of possible unphysically strained structures. Then (3) some C atoms are replaced by O. Finally, (4a) large molecules are screened via a ML approach whereas (4b) smaller molecules are screened with low- and high-quality DFT runs. This way, the authors provide a tool to solve an important problem, that is the systematic exploration of the conformational space of large molecules, such as those contained in biomass (There are however alternatives that still work, without being that extensive and systematic). Further analyses were done via a PCA to provide a rough explanation about how the chemical space looks like.

I have found some results quite impressive and unexpected. For instance, the fact that they described hypervalent binding modes. For instance, CH₃ has been experimentally found to bind to threefold sites on Cu(111) and perhaps other metals. Thus, the C there is strangely coordinated with 6 atoms. This is a well-established result, that also agrees with theoretical simulations. Yet, all other systematic exploration methods I have found out there have been unable to reproduce this.

Besides, the authors walked an extra mile when they rationalized their results by helping themselves with a PCA. The chemical space of the metals under study forms clusters that can be related to the number of bonds between the adsorbates and the surface.

Response: We are grateful to the reviewer for positive feedback and for providing insightful comments that indeed improved the quality of the manuscript.

Comment 2: I have found the methodology rather robust. As Nature Communications focuses on a broad set of readers, the authors need to walk yet another extra mile to assure it is understandable for such an audience. Most of my comments go in this direction. Besides, the authors made available their results, but not the code used to reach such results. I think the Manuscript would only fit Nature Communications tight standards if the code is made fully available (and submitted to peer review in the next round) and a few use examples are included. Otherwise, the paper would better fit Scientific Reports.

Response: We agree with the reviewer and made the code available on https://github.com/VlachosGroup/AdsorptionConfiguration_MS2021.

Comment 3: Regarding the "hypervalent" adsorption states. The authors may extend this discussion and, especially, cite experimental studies. They need to showcase this is a well-established result rather than a weirdness of DFT simulations (that is what my coworkers and I thought the first time we saw that, some years ago). Currently, only a theoretical study is cited.

Response: We thank the reviewer for this suggestion. We cite experimental evidence on page 11, "In this regard, several theoretical and experimental investigations reported that the methyl radical on hollow sites makes three adsorbate-surface bonds (3) and exceeds its valency (1)³⁴⁻³⁶".

Comment 4: The method should also work for molecules containing N, S, or other heteroatoms. Such molecules are important too in the pharmaceutical industry. The authors may mention it.

Response: We thank the reviewer for the suggestion. We added a sentence on page 14 “Also, other heteroatoms such as nitrogen and sulfur with pharmaceutical applications can be considered with trivial extension.”

Comment 5: There are features that make this paper unnecessary complex to understand unless it is re-read several times. For instance, the procedure (which is already complex) is not stated systematically. It is rather outlined three times providing complementary information in lines 62-77, 79-114, 115-147. Such paragraphs also contain many lateral notes, dangling references, and redundancies that drives the reader's mind out of the context, such as:

* "see below" (all instances)

* line 102 "modified force field" (which modifications? it appears just a common quadratic FF).

* Line 146 "enumerating $\times 2$ "

Response: We thank the reviewer for this valuable feedback. To respect the order of the algorithm and serialize the results section, we merged the overview section (lines 79-89) to the paragraph at the end of the introduction and moved the graph enumeration rules section (lines 115 – 147) to beginning of the Results (renamed to Skeleton enumeration). The paragraphs discussing force field and O substitution (lines 90-114) are moved after the Skeleton enumeration and were given their own heading. This simplifies the discussion, and eliminates “see below,” and lateral discussions. We changed the “modified force field” to “the universal force field³² with additional interactions between adsorbate and surface (see method for detail) with heuristic parameters.”

We removed one “enumerating” (which is now on page 5).

Comment 6: In line with the previous comment, each "rule" (or step?) needs to be fully and clearly explained before moving to the next one (ie, merge lines 79-114 and 115-147 into a single story, each paragraph fully explaining one step). The authors mention "two types of rules" out of context in line 137 that are then never more explained. Are they rules by themselves? When is the recursion of lines 132-136 applied?

Response: We agree with the reviewer. The restructuring discussed in the previous comments addresses some of these issues. Regarding the two types of rules, type 1 is for non-surface bonding atoms, and type 2 is for rings, which are both discussed in the same paragraph.

These two rules also add one carbon to the configuration, thus they are executed in recursion. To clarify this, we introduce the recursion process first, and we categorize these rules into 4 more clearly defined types: (1) the addition of an adsorbed carbon to an adsorbed carbon, (2) the addition of non-surface-bonding carbon to a carbon (vacuum propagation rules), (3) the addition of an adsorbed carbon to non-surface-bonding carbon (anchoring rules), and (4) the addition of an adsorbed carbon to two adsorbed carbons forming a ring (ring rules). We explain each type with a dedicated paragraph. These editorial changes do not affect the algorithm but improve clarity.

Comment 7: Also in line with the previous comments, the procedure in Figure 1b needs to be integrated better with the text and Figure 1a. For instance, it is not clear when high-quality DFT is applied.

Response: We agree with the reviewer. We removed Figure 1a entirely as it adds unnecessary complexity, and Figure 1b has been redrawn to improve clarity.

Comment 8: I have read 7+ times lines 125-129 and still do not understand rule 4. Did the authors mean that they reduced their set to structural graphs that are equivalent by symmetry? The full paragraph needs to be rewritten.

Response: We agree that the discussion related to lines 125-129 was not clear, thus we have rewritten the paragraph to “4. Convert the two atom configurations (e.g., green box in inset 2 in **Error! Reference source not found.**a) to graph transformation rules (e.g., blue box in inset 2 of **Error! Reference source not found.**a). A rule consists of a pattern graph (left-hand side of the blue box) and a replacement graph (right-hand side of the blue box). A graph transformation is applied to a configuration by searching for an occurrence of the pattern graph in the configuration, and by replacing the found occurrence with the replacement graph. The two-atom configuration (the green box) becomes the replacement graph (right side of the blue box). The pattern graph (left side of the blue box) is made by removing an atom in two-atom-configurations (green box).”

Comment 9: How did the authors check that the result of the relaxation was OK? Did they check a connectivity matrix, for instance? To be more specific:

Response: We agree that the divergence determination is not clear. We determined the connectivity using the covalent radii. We added “The configuration of the DFT calculated structures is built by determining connectivity between atoms using $d_{ij} < t(r_{cov,i} + r_{cov,j})$ where d_{ij} is the distance between atoms i and j , t is the tolerance factor (1.18 used), and $r_{cov,i}$ is the covalent radius of atom i ” on page 7.

Comment 10: There are structures very relevant for catalysis that are metastable structures (eg, OCCO). When one relaxes such structures, they may either break (eg, to form CO+CO), or tautomerize. Tautomerization is exponentially common on "large" molecules, containing at least 4 C+O atoms. Also, rearrangements may fully change the backbone, thus transforming a "linear" initial guess into a "cyclic" molecule, or even fully moving a functional group to another atom. So, how can they assess that two particular conformations do belong to the same molecule?

Response: Our algorithm enumerates all possible configurations, (e.g., OCCO, and CO; linear backbone and cyclic backbone). The DFT relaxation is performed for all these configurations, thus if the configuration of the OCCO diverges to CO+CO, we know that OCCO is unstable. We also would know that CO is stable as we perform the DFT relaxation for CO as well. If the two tautomers are both stable structures, our algorithm will enumerate both configurations, followed by an assessment of their stability using DFT calculations.

Comment 11: The authors need to comment on how many of such skeletons were branched or

cyclic. Also, how did they assess cycles of three atoms? From Equation 1 and Table 2, it appears that 3-atoms rings (angles around 60°) would all be deemed unstable.

Response: We find that 229 structures out of 4,979 DFT relaxed structures were cyclic, which are all 3 atom-long rings. Indeed, these structures are highly strained and energetically unstable but do not spontaneously dissociate. The 3 atom-long rings do not dissociate from the force-field optimized structures as well.

Comment 12: How did they deal with cis-trans isomerism?

Response: We focused on describing the adsorbates' connectivity pattern to the surface, which is the important first challenge that needs to be addressed for enumeration. To some extent, the cis and trans isomerism is accounted implicitly due to the adsorbates' connectivity pattern as shown below on the left. However, our mechanism does not account for isomerism given the same connectivity pattern as shown below on the right. We added a sentence related to this discussion in the limitations section, "As we focused on the enumeration of adsorbates' connectivity pattern, our scheme does not account for the cis, trans isomerism that is not implicitly accounted by the connectivity pattern (Figure S4)."

implicitly accounted via
surface connectivity

unaccounted

Comment 13: Most importantly, if the input data for the logistic regression is not even partially trained with large molecules (let us say, 5% or 10% of them), how can they describe faithfully the aforementioned complexity sources, especially the tautomerizations?

Response: We had the same concern, thus we chose simple subgraph descriptors, and a simple, logistic regression model to reduce overfitting and enhance extrapolative ability. We tested our model by training it with smaller molecules (≤ 2 C, O atoms) and evaluating it with larger molecules (3 C, O atoms), demonstrating decent extrapolative ability. As our model uses the subgraph descriptors, the idea is that the subgraphs that cause instability in small configurations should also cause instability for large configurations. Especially, the first neighbor effect is likely to be the most important factor, thus we noted in the discussion of the original manuscript that the addition of the 4 heteroatom configuration data will significantly improve our model accuracy.

To clarify this point, we edited page 7, "As the model will be used to predict configurations with larger adsorbates, we devise a similar extrapolation test, where we train the model with adsorbates of ≤ 2 heteroatoms and assess its error on adsorbates with 3 heteroatoms".

In response to comment 4 of Reviewer 1, we performed an additional 1,650 DFT calculations for configurations with 4, 5, and 6 heteroatoms and compared the DFT and FLDLR calculated stability scores. See response and action above.

Comment 14: The very word "rules". "steps" seems to fit the context much better.

Response: We believe that the numbered list in the graph enumeration rules has caused this confusion. We now refer to this numbered list as steps. The rules specifically point to the graph transformation rules, which is a standard term in the graph theory community, while the steps indicate the procedure for systematically constructing the rules.

Comment 15: "enumeration", in "graph enumeration rules". Perhaps "amplification" or "systematic construction" would be more descriptive.

Response: We agree with the reviewer, and changed the graph enumeration rules to graph transformation rules, as discussed in the previous comment.

Comment 16: "Machine Learning"/"ML". As a general criticism to our field: "ML" is becoming a wildcard/generic word, often being used to avoid explaining relevant methodological details or even hiding profound inconsistencies. The authors need to escape such a trap and be crystal clear about their true "ML" core. The fingerprint-like descriptor and how they coupled it with the logistic regression must be fully and dully explained in its corresponding context, even if part of it is fully explained in a previous study. (Are the "fingerprints" equivalent to the "rules" or where are they?). They need a name for the method (eg FLDLR, from "fingerprint-like ...", line 70?) and mention that with a clear, consistent label every time they did use it.

Related to the previous point, PCA is sometimes used as ML, but here the authors used it to get insights, but not as part of their main method. Thus, replacing all instances of "ML" with the name of the specific method used in each step is a must.

Response : We agree with the reviewer and explained our model, and abbreviated our method as FLDLR. We added a paragraph, "To rapidly screen the stability of the configurations with >3 heteroatoms, we develop a fingerprint-like descriptor-based logistic regression (FLDLR), as shown in Figure 3a. Our model input feature is based on fingerprint-like descriptors.³³ In this method, all possible subgraphs of the adsorbate are enumerated, and, for each subgraph, surface atoms connected to the adsorbate are added. The output feature vector contains the number of occurrences for each fingerprint." on page 7.

Comment 17: "Heteroatom". The meaning of "heteroatom" in our specialized field (an atom that is not hydrogen, mostly used in theoretical heterogeneous catalysis) differs from that of IUPAC (an atom that is not C or H, mostly used in organic chemistry). By the context, I am not fully sure which one the authors wanted to convey. The authors would better state the intended meaning in the first use (introduction, line 47).

Response: We agree with the reviewer and clarified the term when first used in the introduction.

Comment 18: "Multi-fidelity DFT". The meaning of this becomes clear once one has read 60% of the manuscript, but not the first times it appears.

Response: We agree with the reviewer. We have improved our discussion as mentioned above, and "multi-fidelity" is now explained when first mentioned in the results section.

Comment 18: "Manifold". A "topological space that locally resembles Euclidean space near each

point" does not seem to fit in the contexts such word was used. Do the authors mean "denticity" of the surface?

Response: We agree with the reviewer, and change the term to “the total number of adsorbate-surface bonds.”

Comment 19: "Valency" (valence), especially in the context of Figure 5, it is more of a "coordination" or "maximum coordination" or "degree of unsaturation". See again IUPAC gold book.

Response: According to the IUPAC gold book, valence is “The maximum number of univalent atoms (originally hydrogen or chlorine atoms) that may combine with an atom of the element under consideration, or with a fragment, or for which an atom of this element can be substituted.” which is in-line with our definition. It also has been used throughout the literature. (*Phys. Rev. Lett.* **2007**, 99, 016105; *Nat. Chem.* **2015**, 7, 403). We apologize if we missed something.

Comment 20: line 32: activation energies are not thermodynamic parameters, but kinetic parameters.

Response: We meant that the number of thermodynamic parameters affecting the kinetics is larger than the number of activation energy parameters affecting it. We modified the sentence “... correlated uncertainty quantification reveals that more thermodynamic parameters than activation energies affect the kinetics.”

Comment 21: line 33: Adsorbate configurations are "prerequisites to compute activation energies". It may read better than "initial or final states in computing activation energies"

Response: We agree with the reviewer and changed the sentence as suggested.

Comment 22: line 43: In this Reviewer's opinion, the challenge is not only the computational cost but the quality of the data obtained when automating. As the volume of results is too high, if the results are not curated by a human, there will always be hidden major problems, not envisaged by the ones doing the automation.

Response: We agree with the reviewer that the quality of the data is very important. In this work, we focused on developing methods to enumerate the configurational space of the adsorbates and reducing the space to a manageable count which has been the critical challenge. We believe that future work could include the curation of the data, which can include human curation, and usage of statistics to identify faulty data. We added a brief comment on this on page 15 “The assessment of the quality of the data is critical. While we addressed the challenge of the enumeration of connectivity patterns, future work should include the curation of the data, which can include manual curation, and the use of statistics to identify faulty data.”

Comment 23: lines 63 and 320, to be replaced by "closed-paced".

Response: We are not aware of the use of the “closed-paced surface” in the field, while the “close-packed surface” has been a standard term. We find that we sometimes used “closely packed,” which were changed to “close-packed.”

Comment 24: line 134: Point 2 unclear: Is the 6-atom backbone reduced to pairs of atoms?

Response: Yes. Any backbones can be reduced to the pairs of atoms in Figure 2, as we consider all possible pair of atoms through systematic enumeration of configurations. We added a sentence on page 5 to further clarify “(e.g., a 6-atom skeleton can be decomposed to the two-atom configurations)”.

Comment 25: line 150-152: unclear sentence. Did it mean "As a configuration is built, occupied surface atoms are not considered for the next iteration"?

Response: The enumeration is continuously performed to an adsorbate on a large lattice without pruning the surface atoms. Surface atoms are pruned for each configuration after the enumeration. We clarify this sentence “After the enumeration, surface atoms in each enumerated configuration are systematically pruned to build a unique, unambiguous graph (see **Error! Reference source not found.**)”.

Comment 26: Figure 2a, inset 3, is very unclear. The text explaining it is also unclear.

Response: We have clarified the inset 3 in step 4 of the skeleton rules section in response to the comment made above.

Comment 27: Figure 2c: What is the physical meaning of the regression parameters 2.13 and -0.99? It is deeply unclear how it can be related to stability.

Response: This line is the decision boundary for the stability of the arc above which “arcs” are stable. $d_{\text{surface}}/d_{\text{nearest_neighbor}}$ indicates the normalized distance along the surface, thus the longer the normalized distance, the more atoms the arc needs, which, otherwise, will be strained and unstable. We added the following line on page 5: “The line between the stable and unstable data indicates the decision boundary we used to decide the arcs’ stability.”

Comment 28: line 175: please define "hyperparameter" for a broad audience.

Response: Instead of using “hyperparameter”, we edited the sentence to make it clearer to a broader audience: “The probability threshold is used as a tunable parameter for screening.”

Comment 29: line 183: why the harmonic mean? why not the geometric mean, or even a fully new descriptor with a clear interpretation such as $(TP+TN)/(TP+TN+FP+FN)$.

Response: In the binary classification, the harmonic mean of precision and recall has been used as the balanced mean of the two, defined as the F1 score. Besides theoretical reasons, the practical justification is that the harmonic mean is more conservative than geometric mean, i.e., both precision and recall have to be high to get a high F1 score, more so than the geometric mean.

Comment 30: Figure 3 b-e: The terms in the "y" axis are very hard to understand. A clear definition (equation) of each quantity may be provided as insets. Besides, the methods used to get the 95% and 99% must be fully explained.

Response: We agree with the reviewer. We added the metrics as an inset and a sentence to the caption, “The inset indicates the metric equation where T, F, P, and N indicate true, false, positive,

and negative, respectively. The threshold is a tunable probability decision boundary to predict stability, which results in different performances for the metrics considered in this study.”

Comment 31: line 228: The number of adsorbate-surface bonds and the d-band centre have been previously linked via a PCA, Nat. Commun 2019, 10, 4687, with rather similar conclusions.

Response: We thank the reviewer for this oversight. We added “Similarly, the d-band center has also been shown to correlate to the energy of adsorbates with a varying number of adsorbate-surface bonds.³⁷”

Comment 32: line 233-251 and Figure 5c: It is not fully clear what is the difference between both PCA. The authors label them "adsorbable molecules" and mention ethylene as an example. Did they mean "closed-shell molecules"?

Response: We agree with the reviewer the difference is not clear. We meant any molecules considered in this study. We changed the “adsorbable molecule” in Figure 5c to “molecule adsorption stability matrix” which is defined in more detail in the main text. The difference between the two PCA input matrices is that the one runs over the configurations while the other runs over the molecules (e.g. a molecule is stable if any adsorption configuration of it is stable). We clarify this by adding a sentence on page 10, “Compared to the previous matrix, the second dimension runs over molecules.” We also added reference to the “molecule adsorption stability matrix” in the main text so that readers can find its mathematical definition.

Comment 33: line 244: I understand that closed-shell molecules do not adsorb on too noble surfaces such as Au. Yet, I wonder if the CH₂C is a typo (meaning HCCH), or if the authors consider it an "adsorbable molecule" despite being a (probably unstable) biradical in gas-phase.

Response: We did not mean the listed molecules are stable in gas-phase but are reaction intermediates on the catalytic surface. We changed the term, “adsorbable molecule,” in the previous comment, thus it should be clear now.

Comment 34: Figure 5b: The "intuitive" word is tricky in this context ($n_{\text{bonds}} \approx \text{valence/coordination}$) and I suggest the authors remove it. For instance, my intuition says that acetaldehyde would "physisorb" ($0 = n_{\text{bonds}} < \text{valence} = 2$) rather than "chemisorb" with C-surface and O-surface bonds, for maybe all metals ($n_{\text{bonds}} = \text{valence} = 2$).

Response: We agree with the reviewer and removed the word “intuitive” in Figure 5b.

Comment 35: Related to the previous one, how is that formic acid is "unstable" on Au?

Response: We find that formic acid chemisorbs on 9 metals (Ru0001, Ni111, Pt111, Co0001, Ir111, Ag111, Re0001, Rh111, and Cu111), but not on Au.

Comment 36: line 277: the word "mechanism" in this context should instead be "path". In a complex reaction network, a "mechanism" is a collection of overlapping "paths". It is true that in old-school mechanistic studies, due to the lack of computer power, only a particular path was assumed for all materials being screened. However, this is seldom done in serious studies nowadays.

Response: We agree with the reviewer and changed the “mechanism” to “reaction pathway.”

Comment 37: line 282: The authors need to explain what a BEP relation is for a broad audience. I would rather avoid that and would mention how they would translate this method to find transition states (which is 100× more interesting than a BEP relationship!).

Response: We agree with the reviewer. We explained the BEP relation by adding “(scaling relationship between reaction energy and activation energy),” and also added transition state structure as an example.

Comment 38: line 324, when doing the low-fi DFT at 300 eV, which pseudopotentials did the authors use for C and O? the "normal" ones, or the C_s O_s version?

Response: We used C_h, H_h, O_h versions of the pseudopotential as recommended in the VASP website for the molecules with shorter bonds.

Comment 39: line 339: To which molecule are these "local ground states" related?

Response: These include various molecules, and they are local ground states with respect to each molecule’s global minimum configuration. We changed the sentence to “Four configurations are local ground states (0.13, 0.19, 0.12, and 0.01 eV with respect to each molecule’s ground state configuration).”

Reviewer 3

Comment: The manuscript presents the results of a configurational search scheme for organics on metal surfaces employing graph representations of configurations, force field optimization, DFT simulations and machine learning. The authors claim to be able to identify the optimal binding configuration of adsorbates of up to 6 C/O atoms. The strength of this manuscript are:
A, This is a significant improvement over similar attempts particularly, ref 27 which can search very limited configurational spaces, and ref 29 and 30 which are similar in spirit but are case specific as opposed to general.

B, The presentation of the algorithm is, for the most part, very clear and well documented.

C, Its potential use in identifying intermediates along a catalytic pathway will be highly valuable to the catalyst community at large.

Response: We are grateful to the reviewer for the positive feedback and suggestions.

Comment 1: Nonetheless I have some difficulties with this work.
A, It seems to have very unmeritable scaling to the point where systems with 10 atoms would require 10⁵-10⁶ DFT simulations which is impractical. Yet many many biomasses derived molecules will be in this size range or larger. For a first iteration of an algorithm this limitation is to be expected but the authors should admit this limitation and at least discuss potential algorithmic advances to improve upon this to reach large scaling.

Response: We agree with the reviewer that this limitation has not been discussed. We added a paragraph and a potential remedy at the end of the discussion. See also comment 3 of Reviewer 1 where this one and other limitations are discussed.

Comment 2: B, Along these lines the authors have avoided references to global optimization techniques-which is in a sense what they are proposing. A few notable algorithms even claim to be able to scale to large dimensional systems using adaptive learning

Ex

1, Zhang, et al (2020). NWPEsSe: an adaptive-learning global optimization algorithm for nanosized cluster systems. Journal of chemical theory and computation, 16(6), 3947-3958.

2, Janet, et al (2020). Accurate multiobjective design in a space of millions of transition metal complexes with neural-network-driven efficient global optimization. ACS central science, 6(4), 513-524.

3, Bisbo, M. K., et al. (2020). Efficient global structure optimization with a machine-learned surrogate model. Physical review letters, 124(8), 086102.
Approaches such as these may be beneficial to hybridize into the current model to further reduce the number for DFT simulations.

Response: We are grateful for the suggestion to improve the scalability. We discuss now this point in the limitations section that cite the suggested references, “Our scheme shares similarities with global optimization techniques which the aim to identify all local and global minima in a high-dimensional space. Integration with these advanced global optimization algorithms⁴⁸⁻⁵⁰ is expected to improve scalability.” See also comment 3 of Reviewer 1 for more expanded response and action.

Comment 3: C, The utility and strength of the approach would be best served by presenting an illustrative example of catalyst discovery-ex epoxidation reactions on Ag etc. As it is now, its just a complex configuration space sampling tool and, although interesting, potentially to specialized for nature Comm. However, I do believe the authors can rectify this.

Response: Thanks for the valuable suggestion. This was also suggested by Reviewer 1, comment 4, which we addressed above and revised the manuscript accordingly.

REVIEWERS' COMMENTS

Reviewer #1 (Remarks to the Author):

The authors have addressed most of my concerns. The paper is now significantly improved, and I recommend its publication in Nature Communications. I suggest the authors to consider the following optional changes.

Previously, I have asked the authors to clarify how the stability is quantified. They explained that "the stability is quantified as 1 (stable) or 0 (unstable)". This is a step in the right direction, but it is still unclear which exactly number is used to decide between 0 and 1. Is it the magnitude of adsorption energy? I think it is important to give a formula that is used to calculate stability indicator.

Reviewer #2 (Remarks to the Author):

Gu, Lee, Jung, and Vlachos replied adequately to the points raised by the Reviewers, greatly improving the readability, aesthetics, and potential impact of the Manuscript. The code has been made open. I see no holes in the methods or results that may hinder reproducibility or invalidate the claims. As I stated in my initial report, I have found some of their results quite impressive. I hereby recommend publication in its current form.

Reviewer #3 (Remarks to the Author):

The revisions to the manuscript are acceptable.

Point-by-point responses and new text in blue.

Reviewer 1

Comment 1: The authors have addressed most of my concerns. The paper is now significantly improved, and I recommend its publication in Nature Communications. I suggest the authors to consider the following optional changes.

Response: We are grateful to the reviewer for finding our paper significantly improved and suitable for publication in Nature Communication.

Comment 2: Previously, I have asked the authors to clarify how the stability is quantified. They explained that "the stability is quantified as 1 (stable) or 0 (unstable)". This is a step in the right direction, but it is still unclear which exactly number is used to decide between 0 and 1. Is it the magnitude of adsorption energy? I think it is important to give a formula that is used to calculate stability indicator.

Response: We agree with the reviewer that the method for labeling configuration stability is not clear. We labeled a configuration as stable if the connectivity does not change after the DFT relaxation of the structure (i.e. the configuration represents a local or global minimum on the potential surface). If the connectivity pattern changes upon DFT relaxation, we labeled them unstable, as the configuration represents an unstable point on the potential surface.

We added the following sentence on page 7 after the "... the stability is quantified as 1 (stable) or 0 (unstable)."

"A configuration is labeled stable if the connectivity does not change after the DFT relaxation (i.e. the configuration represents a local or global minimum on the potential surface). If the connectivity pattern changes upon DFT relaxation, we labeled them unstable, as the configuration represents an unstable point on the potential surface."

Reviewer 2

Comment 1: Gu, Lee, Jung, and Vlachos replied adequately to the points raised by the Reviewers, greatly improving the readability, aesthetics, and potential impact of the Manuscript. The code has been made open. I see no holes in the methods or results that may hinder reproducibility or invalidate the claims. As I stated in my initial report, I have found some of their results quite impressive. I hereby recommend publication in its current form.

Response: We are grateful to the reviewer for providing insightful comments that indeed improved the quality of the manuscript.

Reviewer 3

Comment 1: The revisions to the manuscript are acceptable.

Response: We thank the reviewer for finding the paper acceptable for publication.